# ML-Guided Primal Heuristics for Mixed Binary Quadratic Programs

## Abstract

Mixed Binary Quadratic Programs (MBQPs) are an important and complex set of problems in combinatorial optimization. As solving large-scale combinatorial optimization problems is challenging, primal heuristics have been developed to quickly identify high-quality solutions within a short amount of time. Recently, a growing body of research has also used machine learning to accelerate solution methods for challenging combinatorial optimization problems. Despite the increasing popularity of these ML-guided methods, a large body of work has focused on Mixed-Integer Linear Programs (MILPs). MBQPs are challenging to solve due to the combinatorial complexity coupled with nonlinearities. This work proposes ML-guided primal heuristics for Mixed Binary Quadratic Programs (MBQPs) by adapting and extending existing work on ML-guided MILP solution prediction to MBQPs. We introduce a new neural network architecture for MBQP solution prediction and a new training data collection procedure. Moreover, we extend existing loss functions in solution prediction and propose to combine contrastive weighted cross-entropy losses. We evaluate the methods on standard and real-world MBQP benchmarks and show that the developed ML-guided methods significantly outperform existing primal heuristics and state-of-the-art solvers. Furthermore, models trained with our proposed extension with combined losses outperform other ML-based methods adapted from MILPs and improve generalization in cross-regional inference on a real-world wind farm layout optimization problem.

## 1 Introduction

Mixed Binary Quadratic Programs (MBQPs) are discrete optimization problems with quadratic terms in the objective function subject to a set of linear constraints. MBQPs encode many important problems in Combinatorial Optimization (CO) (Loiola et al., 2007; Rebennack, 2024; Kochenberger et al., 2005) and cover a wide range of applications, including finance (Parpas & Rustem, 2006), machine learning (Bertsimas & Shioda, 2009), as well as chemical (Misener & Floudas, 2013) and energy systems (Turner et al., 2014). A significant body of research on CO algorithms has focused on *primal heuristics*, which are algorithms designed to find good feasible solutions quickly and without optimality guarantees (Berthold, 2014).

Despite development in solvers and heuristics, solving large-scale COs remains challenging. In recent years, Machine Learning (ML) has been proposed to accelerate solution methods for CO problems. Motivated by the fact that CO problems sharing similar structures are solved repeatedly in many applications (Huang et al., 2024b; Scavuzzo et al., 2024), a growing body of research uses ML to guide algorithmic policies or to build new policies customized to instances that appear in specific applications. For example, Han et al. (2023); Nair et al. (2020); Huang et al. (2024a); Ding et al. (2020) proposed ML-guided primal heuristics for Mixed Integer Linear Programs (MILPs), wherein they they predict the optimal assignment for a subset of the variables. While prior work on ML-guided CO methods has shown success across multiple algorithmic components on many challenging CO problems, existing work in this area has mainly focused on MILPs. A small body of research has used ML to advance solution methods for general nonlinear programming problems (Bonami et al., 2018; Bagga & Delarue, 2023; Ghaddar et al., 2023; Ferber et al., 2023; Tang et al., 2025; Ferber et al., 2023), but ML-guided methods in this space are not as well developed as in MILPs.

MBQPs are even more challenging to solve than MILPs due to the combinatorial nature (Magnanti, 1981) coupled with nonlinearities. In this work, we develop ML-guided primal heuristics for MBQPs by adapting and extending existing work on ML-guided MILP solution methods. We adapt the Weighted Cross-Entropy-based and Contrastive Learning-based methods which are used in MILP solution prediction to MBQPs. To adapt to MBQPs, we propose a novel neural network architecture that extracts input features and a new data collection procedure that generates high-quality solutions as ground truth training data for large-scale MBQPs. Furthermore, we extend existing loss functions used in CO solution prediction and propose to combine Cross-Entropy and and Contrastive losses. Computational results show that the adapted and extended ML methods outperform existing primal heuristics and state-of-the-art solvers on standard and real-world MBQP benchmarks. Furthermore, we show that solution prediction models trained with our proposed extended loss function outperform other ML-guided methods adapted from MILPs in in-domain testing and improves generalization in cross-regional inference on a real-world wind farm layout optimization problem.

## 2 BACKGROUND AND RELATED WORKS

### 2.1 MIXED BINARY QUADRATIC PROGRAMS

A Mixed Binary Quadratic Program (MBQP) with $n$ decision variables is defined as

$$\min x^T H x + c^T x \quad \text{s.t. } Ax \leq b \text{ and } x_j \in \{0, 1\}, \forall j \in B \tag{1}$$

where $H \in \mathbb{R}^{n \times n}$, $c \in \mathbb{R}^n$, $A \in \mathbb{R}^{m \times n}$, and $b \in \mathbb{R}^m$. $H$ is a real symmetric matrix that encodes quadratic terms in the objective function and is not necessarily positive semidefinite, allowing for nonconvex objective functions. $B \subseteq \{1, ..., n\}$ is the set of binary decision variables.

**Solution methods** MBQPs are NP-hard in general (Pia et al., 2017). The Branch-and-Bound (BnB) algorithm is an exact tree search algorithm to solve MILPs, MBQPs and more general Mixed-Integer Nonlinear Programming (MINLP) problems. As large-scale MBQPs are challenging to solve with exact methods, a significant body of research has focused on *primal heuristics*, which are algorithms designed to quickly identify high-quality feasible solutions for a given optimization problem without optimality guarantees (Berthold, 2014). These heuristics typically involve solving a relaxation of the original problem and then creating a subproblem by fixing a subset of integer variables by rounding the relaxation values to the nearest integer values, such as RENS (Berthold, 2014), Undercover (Berthold & Gleixner, 2014), and Relax-Search (Huang et al., 2025).

### 2.2 SOLUTION PREDICTION FOR MILPs

Previous work on using ML to accelerate solving CO problems has been focused on Mixed Integer Linear Programming (MILP). An MILP can be viewed as the subclass of MBQPs in Eqn. 1 where the quadratic term matrix $H$ is the zero matrix. The goal of an MILP is to find $x$ such that $c^T x$ is minimized, subject to $Ax \leq b$ and integrality constraints $x_j \in \{0, 1\}, \forall j \in B$. A large body of ML-guided primal heuristics for MILPs are based on predicting partial solutions (Nair et al., 2020; Han et al., 2023; Huang et al., 2024a; Ding et al., 2020).

**Solution prediction** Nair et al. (2020) and Han et al. (2023) use Weighted Cross-Entropy (WCE) loss to learn the probability distribution of the solution space of an MILP instance $M$. The goal is to learn from a set of multiple solutions, weighted by the quality of the solution. Specifically, for a solution $\boldsymbol{x}$, the energy function $E(\boldsymbol{x}; M)$ is defined as $c^T \boldsymbol{x}$ if $\boldsymbol{x}$ is feasible, or $\infty$ otherwise, assuming minimization. Given $M$, the conditional distribution of a solution $\boldsymbol{x}$ is modeled as

$$P(x|M) \equiv \frac{\exp(-E(x; M))}{\sum_{x'} \exp(-E(x'; M))} \tag{2}$$

, so that solutions with better objective values have higher probability. The learning task is to train a model $p(x|\theta, M)$ parameterized by $\theta$ that approximates $p(x|M)$. To collect training data, Nair et al. (2020) and Han et al. (2023) obtain the set of solutions by running state-of-the-art MILP solvers for a large amount of time. Instead of using WCE loss, Huang et al. (2024a) learn $p(x|\theta, M)$ using Contrastive Learning (CL). The CL-based method makes discriminative predictions by contrasting

the positive samples (i.e., good solutions) and negative samples (i.e., bad solutions). Positive samples are obtained by running MILP solvers, similar to (Nair et al., 2020) and (Han et al., 2023). Negative samples are obtained by solving another MILP that searches for bad variable assignments within some Hamming distance of the good solutions.

**Inference** Since the full prediction might not be feasible, ML-guided primal heuristics for MILPs involve solving another MILP at inference time. Nair et al. (2020) use Neural Diving (ND), which uses the prediction of a subset of the variables and creates a smaller sub-MILP that is easier to solve after fixing the subset. The size of this sub-MILP is controlled by the ratio of variables that are fixed. Han et al. (2023) and Huang et al. (2024a) use a Predict-and-Search (PaS) framework that searches for feasible solutions within some neighborhood of the full prediction by adding a cut to the original MILP. The degrees of freedom in PaS are controlled by the number of variables that are allowed to be different from the prediction. The ND approach allows for faster runtime at inference time as the subproblem contains a small number of variables, but the solutions returned can be more suboptimal. PAS has more freedom to correct errors from the ML predictions, but can be harder to optimize because the size of the MILP at inference contains the same number of variables as the original MILP.

## 2.3 ML-GUIDED METHODS FOR NONLINEAR OPTIMIZATION

There has also been research on ML for general nonlinear optimization. A large body of research in this space focuses on continuous problems. For example, Ghaddar et al. (2023) uses ML to select branching rules in polynomial optimization. Liang & Chen (2024) learn solution mapping for nonconvex optimization using RectFlow (Liu et al., 2022). Baltean-Lugojan et al. (2019) uses ML to score semidefinite cutting planes in linear outer-approximations of non-convex quadratic programming problems with box constraints. Saravanos et al. (2025) develops an ML-aided distributed optimization technique for large-scale quadratic programming problems. Gao et al. (2024) develops a learning-based interior point method for solving continuous constrained nonlinear programs. For general MINLPs, Tang et al. (2025); Ferber et al. (2023) develop a gradient-based methods using self-supervised learning and projection. There has also been learning-based methods for specific applications of MBQPs such as the quadratic assignment problem (Bagga & Delarue, 2023) and the unit commitment problem (Maisonneuve & Lesage-Landry, 2024). To our knowledge, there has been limited existing work on ML-guided methods for solving general MBQPs. Bonami et al. (2018) uses ML to decide on whether to linearize the quadratic terms in the objective function or not. Chen et al. (2024) studies the expressiveness of graph neural networks for continuous and mixed-integer quadratic programs.

## 3 METHODS

We develop an ML-guided primal heuristic for MBQPs based on solution prediction, as shown in Fig. 1. An input MBQP is represented as a tripartite graph (Fig. 1 (B)) and then passed to a Graph Attention Network module (Fig. 1 (C)) which produces solution predictions for binary decision variables in MBQPs. At inference time, the predicted solutions are used to create a sub-MBQP (Fig. 1 (F)). We introduce a new method for collecting training data for MBQPs (Fig. 1 (D)). In training the models, we adapt the WCE and CL losses which have been used in solution prediction in MILPs to MBQPs and propose an extended loss function that combines CL and WCE losses (Fig. 1 (E)).

## 3.1 NEURAL NETWORK ARCHITECTURE

**Tripartite graph representation** We propose a tripartite graph representation of MBQP instances (Fig. 1 (B)). The tripartite graph contains three sets of nodes: the constraint nodes ($C$), variable nodes ($V$), and quadratic term nodes ($Q$). A $C - V$ edge connects a variable and a constraint if the variable has a non-zero coefficient in the constraint. A $Q - V$ edge connects two $V$ nodes if the two variables appear in the same quadratic term. The sets of features in the $C$ and $V$ nodes are adapted from solution prediction for MILPs in (Han et al., 2023). For the $Q$ nodes, we propose a custom feature set that captures the characteristics of the quadratic terms in the objective function of MBQPs. Details on input features are deferred to Appendix F.

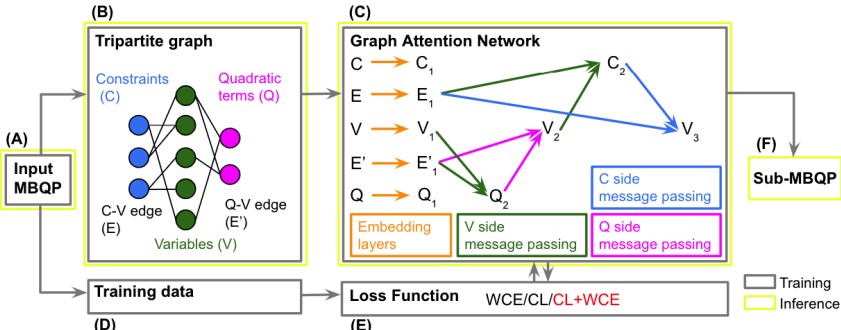

Figure 1: Training/Inference pipeline for ML-guided MBQP solving via solution prediction.

**Graph Attention Network** We learn a policy $p(x|\theta; M)$ parameterized by $\theta$ that processes the featured tripartite graph and outputs predictions of the variables for instance $M$ using a Graph Attention Network (GAT) (Brody et al., 2021). The GAT performs four rounds of message passing, as shown in Fig. 1 (C). In round one, each quadratic term node in $Q_1$ attends over its neighbors in $V_1$ using $H$ attention heads to produce updated quadratic term embeddings $Q_2$. In round two, each variable node in $V_1$ attends over its neighbors in $Q_2$ to produce updated variable embeddings $V_2$. In round three, each constraint node in $C_1$ attends over its neighbors in $V_2$ to produce updated constraint embeddings $C_2$. In the final round, each variable node in $V_2$ attends over its neighbors in $C_2$ to produce the final variable embeddings $V_3$. The message passing outputs are then passed through Multi-Layer Perceptrons (MLPs) followed by activation layers to obtain the final output $p(x|\theta; M)$. Details on the ML architecture are deferred to Appendix F.

## 3.2 LOSS FUNCTION

The solution prediction policy $p(x|\theta; M)$ can be learned with different approaches (Fig. 1 (E)). In this work, we first adapt the WCE and CL losses that have been used for MILPs to MBQPs. Then, we propose an extension that combines CL and WCE losses to improve the performance.

**Weighted Cross-Entropy** Following the Weighted Cross-Entropy (WCE) (Han et al., 2023) approach, we create a training dataset that contains $N$ MBQPs instances $\left\{\left(M_i, S_{M_i}^+\right)\right\}_{i=1}^{N}$, where $S_{M_i}^+$ is a set of unique solutions for the instance $M_i$. Let $p\left(x^+|\theta; M_i\right)$ denote the probability of solution $x^+$ given instance $M_i$ as the input. We adapt the energy function $E(\boldsymbol{x}; M)$ in Eqn. (2) to the case of MBQPs to assign higher probability for better solutions. For a solution $\boldsymbol{x}$, the energy function $E(\boldsymbol{x}; M)$ is defined as $x^T H x + c^T x$ if $\boldsymbol{x}$ is feasible. During training, for instance $M_i$ with quadratic term matrix $H_i$ and cost vector $c_i$, the weight applied to the solution $x_+$ is $w(x^+|M_i) \equiv \frac{\exp\left(-x^{+\top} H_i x^+ - c_i^{\top} x^+\right)}{\sum_{x \in S_{M_i}^+} \exp\left(-x^{\top} H_i x - c_i^{\top} x\right)}$. Based on the Kullback-Leibler divergence which measures the distance between the conditional distribution in Equation 2 and the learned policy, the loss function to be minimized is:

$$\mathcal{L}^{\text{WCE}}(\theta) \equiv -\sum_{i=1}^{N} \sum_{x^+ \in S_{M_i}^+} w(x^+|M_i) \log p\left(x^+|\theta; M_i\right). \tag{3}$$

**Contrastive Learning** Following the CL-based approach (Huang et al., 2024a), let $\left\{\left(S_{M_i}^+, S_{M_i}^-\right)\right\}_{i=1}^{N}$ be a training dataset of $N$ MBQP instances, where $S_{M_i}^+$ and $S_{M_i}^-$ are the sets of positive and negative samples for instance $M_i$, respectively. We use a form of the NT-Xent Loss (Chen et al., 2020) to learn to distinguish between positive and negative samples. We use the the · operator to denote the dot-product similarity. Let $p(\theta; M_i)$ be the predicted solution vector given instance $M_i$ as the input. The loss function to be minimized is

$$\mathcal{L}^{\text{CL}}(\theta) = \sum_{i=1}^{N} \sum_{x^+ \in S_+^{M_i}} \mathcal{L}^+\left(\theta \mid x^+, M_i\right), \tag{4}$$

where

$$\mathcal{L}^+ \left( \theta \mid x^+, M_i \right) = -\log \frac{\exp \left( x^+ \cdot p(\theta; M_i) / \tau(x^+ | M_i) \right)}{\sum_{\tilde{x} \in S_{M_i}^-} \exp \left( \tilde{x} \cdot p(\theta; M_i) / \tau(x^+ | M_i) \right)}. \tag{5}$$

Based on the dot-product similarity, the loss value $\mathcal{L}^+ \left( \theta \mid x^+, M_i \right)$ is low when $p(\theta; M_i)$ is similar to the positive sample $x^+$ and dissimilar to negative samples $\tilde{x} \in S_{M_i}^-$. $\tau(x|M_i)$ is a temperature parameter that scales the similarity scores in Eqn. 5. We set $\tau(x|M_i) = \frac{1}{\exp((x^T H_i x + c_i^T x)/w)}$ with $w < 0$ for minimization problems, so that the loss $\mathcal{L}^+ \left( \theta \mid x^+, M_i \right)$ for better $x^+$ (i.e., positive sample with a lower objective value) has a lower temperature parameter, which creates a sharper distribution and increases the penalties on the negative samples.

**Combining Contrastive Learning and Weighted Cross-Entropy**   In addition to extending loss functions used in MILP solution prediction, we propose to combine CL and WCE losses. It has been observed that for each positive sample $x^+$, a subset of variables often have the same assignments across $x^+$ and the corresponding negative samples in the CL-based approach, and that the CL loss $\mathcal{L}^+ \left( \theta \mid x^+, M_i \right)$ in Eqn. 5 does not depend on the predicted solution by the ML model for this subset. This observation is formalized in Proposition 1; the proof is deferred to Appendix B. We also show empirically in Section. 4.2 that the percentage of this subset can be large in MBQP benchmarks.

**Proposition 1** *Given an MBQP instance $M_i$ with sets of positive and negative samples $\left( \mathcal{S}_{M_i}^+, \mathcal{S}_{M_i}^- \right)$, let $\mathcal{B}_i$ be the index set of all binary decision variables. Let $x^+ \in \mathcal{S}_{M_i}^+$ be any positive sample. Let $\mathcal{U}_i^{x^+} \subseteq \mathcal{B}_i$ be an index set such that $\tilde{x}_d = t_d \ \forall d \in \mathcal{U}_i^{x^+} \ \forall \tilde{x} \in S_{M_i}^- \cup \{x^+\}$ where $t_d \in \{0, 1\}$ is the assignment for variable with index $d$. The CL loss $\mathcal{L}^+ \left( \theta \mid x^+, M_i \right)$ in Eqn. 5 depends only on the predictions for the subset of variables in $\mathcal{B}_i \setminus \mathcal{U}_i^{x^+}$.*

As shown in Proposition 1, the CL loss $\mathcal{L}^+ \left( \theta \mid x^+, M_i \right)$ for each positive sample $x^+$ only penalizes predictions that are similar to negative samples for variables that are not in the subset $\mathcal{U}_{x^+}^i$. Therefore, to improve the prediction for variables in $\mathcal{U}_{x^+}^i$, we propose to apply classification of whether the variable takes 1 or 0 as the solution value using binary Cross-Entropy (CE) loss, given that variables in this set takes the same solution value in $S_{M_i}^- \cup \{x^+\}$ by definition. For each positive sample $x^+$, we apply the CL loss in Eqn. 5 (which only applies to $\mathcal{B}_i \setminus \mathcal{U}_{x^+}^i$) and CE loss for $\mathcal{U}_{x^+}^i$. For each instance $M_i$, the CE loss for each $x^+ \in \mathcal{S}_+^{M_i}$ is weighted by the adapted energy function in Eqn. 2 (denoted as $w(x^+ | M_i)$). Formally, for each positive sample $x^+ \in \mathcal{S}_{M_i}^+$, let $t_{d,i}$ denote the assignment of $d^{th}$ variable, which is the same across $S_{M_i}^- \cup \{x^+\}$ for $d \in \mathcal{U}_i^{x^+}$. Let $\hat{p}_{d,i} \equiv p(x_{d,i} = 1 | \theta; M_i)$ be the probability that the $d^{th}$ variable of $M_i$ takes a solution value of 1 predicted by the ML policy. The CE loss for each $x^+$ is $\mathcal{L}^{\mathrm{CE}}(\theta \mid x^+, M_i) = \sum_{d \in \mathcal{U}_i^{x^+}} t_{d,i} \log(\hat{p}_{d,i}) + (1 - t_{d,i}) \log(1 - \hat{p}_{d,i})$. Accounting for the weights for each postive sample, the combined loss function to be minimized is

$$\mathcal{L}^{\mathrm{CL+WCE}}(\theta) = \sum_i^N ( \sum_{x^+ \in \mathcal{S}_{M_i}^+} \lambda^{CL} \mathcal{L}^+ \left( \theta \mid x^+, M_i \right) + w(x^+ | M_i) \mathcal{L}^{\mathrm{CE}}(\theta \mid x^+, M_i)), \tag{6}$$

, where $\lambda^{CL}$ is a hyperparameter that controls the weight of the CL loss compared to WCE loss.

### 3.3   TRAINING DATA COLLECTION FOR MBQPs

Training data collection (Fig. 1 (D)) consists of compiling multiple good solutions (and bad solutions in the CL-based approach) that can be used for solution prediction in MBQPs. We propose *Randomized Relax-Search*, a novel heuristic that produces a set of diverse high-quality solutions for MBQPs. *Randomized Relax-Search* is extended from the *Relax-Search* (Huang et al., 2025) heuristic, which uses a suboptimal relaxation solution of the MBQP as the basis, fixes a subset of variables using the rounded relaxation, and searches over a sub-MBQP. *Randomized Relax-Search* introduces randomization to create $K$ sub-MBQPs, as shown in Algorithm 1. In solving the $k^{th}$ sub-MBQP, the best solution $x_k^+$ and the worst solution $x_k^-$ are stored. The procedure returns the set of best solutions $S^+$ and worst solutions $S^-$ after solving $K$ sub-MBQPs.

---

**Algorithm 1** Randomized Relax-Search for training data collection

---

**Require:** A MBQP $\mathcal{P}$ with set of binary variables $\mathcal{B}$, relaxation time limit $T_r$, subproblem time limit $T_s$, number of random seeds $K$, candidate fixing ratio $p_1$, final fixing ratio $p_2$ ($p_1 > p_2$)
1: Relaxed solutions $\bar{x} \leftarrow$ Compute the Nonlinear Programming relaxation of $\mathcal{P}$ given time limit $T_r$
2: Set of good solutions $S^+ \leftarrow \emptyset$
3: Set of bad solutions $S^- \leftarrow \emptyset$
4: Candidate set $\mathcal{C} \leftarrow$ select $p_1 * |\mathcal{B}|$ variables that are least fractional variables in $\bar{x}$.
5: **for** $k \in 1, 2, ..., K$ **do**
6: $\quad \mathcal{C}'_k \leftarrow$ Randomly and uniformly select $p_2 * |\mathcal{B}|$ variables from $\mathcal{C}$
7: $\quad$ **for** $i \in \mathcal{C}'_k$ **do**
8: $\quad\quad$ Fix $x_i = \lfloor \bar{x}_i \rceil$ by rounding to the nearest integer
9: $\quad$ **end for**
10: $\quad x_k^+, x_k^- \leftarrow$ Best and worst solutions obtained by solving the $k^{th}$ sub-MBQP with a complete solver, given time limit $T_s$.
11: $\quad S^+ \leftarrow S^+ \cup \{x_k^+\}$
12: $\quad S^- \leftarrow S^- \cup \{x_k^-\}$
13: **end for**
14: **return** $S^+, S^-$

---

For training with WCE loss, the set of good solutions $S^+$ is used. For CL losses, $S^+$ is used as the set of positive samples. We denote the worst solution value from $S^+$ as $v' = \max_{x \in S^+} x^T H x + c^T x$. For the set of negative samples in CL, we use $\{x | (x^T H x + c^T x) > v', x \in S^-\}$. In other words, we only include solutions in $S^-$ that have worse objective values than the worst solutions in $S^+$.

### 3.4 INFERENCE

At inference time, we choose to use the ND-based method discussed in Subsection 2.2 which reduces the original problem to a smaller sub-MBQP (Fig. 1) (F), as our goal is to develop fast primal heuristics. The PaS-based method is challenging for MBQPs because it requires solving another MBQP of the same size. After obtaining the variable predictions, we create a sub-MBQP by fixing the top $p$ percent of variables for which the ML model is most confident with (i.e., least fractional in the predictions). The resulting sub-MBQP is then solved with a CO solver.

## 4 COMPUTATIONAL EXPERIMENTS

### 4.1 SETUP

**Benchmarks** We evaluate the methods on synthetic and real-world benchmarks. For sythetic benchmarks, we include the Cardinality-constrained Binary Quadratic Programs (CBQP) (Zheng et al., 2012), Cardinality-constrained Quadratic Knapsack Problem (CQKP) (Létocart et al., 2014), and the Quadratic Multidimensional Knapsack Problem (QMKP) (Forrester & Hunt-Isaak, 2020). All synthetic benchmark instances contain 1000 binary variables and have a quadratic term density of 25%. Moreover, we test on a real-world *Wind Farm Layout Optimization Problem* (WFLOP). WFLOP seeks to identify the placement of a set of wind turbines within a fixed area to maximize power generation across all turbines and over all wind scenarios while also satisfying minimum separation constraints. We use the MBQP formulation of WFLOP in (Huang et al., 2025) and instantiate the instances using probabilistic wind models (i.e., probability density functions sampled from the NOW-23 dataset (Bodini et al., 2023)) which represents long-term wind patterns over a 10 year time horizon at selected locations in the California offshore region. The WFLOP instances contain 1000 binary variables while the quadratic term densities depend on the wind distribution. On average, the WFLOP-California instances have a quadratic term density of 32.42%. Details on the benchmarks are deferred to Appendix A.

**Evaluation Metrics** We use the following metrics to evaluate the effectiveness of different methods: (1) The *Primal Gap* (PG) (Berthold, 2013) is the normalized difference between the objective value $v$ found by a method and a best known objective value $v^*$, defined as $PG = \frac{|v-v^*|}{max(|v|,|v^*|)}$, when

$vv^* > 0$. When no feasible solution is found or when $vv^* < 0$, PG is defined to be 1. PG is 0 when $|v| = |v^*| = 0$. (2) The *Primal Integral* (PI) (Berthold, 2013) is the integral of the primal gap over time, which captures the speed at which better solutions are found. (3) The *# wins* in terms of PI is the number of test instances for which the method results in the lowest PI across all other methods.

**Baselines** We compare the proposed ML-guided MBQP solving methods adapted from MILPs (WCE and CL) and the proposed method with the extended loss function (CL+WCE) with well-established primal heuristics for MBQPs and general MINLPs, including RENS (Berthold, 2014), Undercover (Berthold & Gleixner, 2014), and Relax-Search (Huang et al., 2025). In addition, we compare with the state-of-the-art MINLP solver SCIP (Bestuzheva et al., 2021), which uses BnB as its core component and includes primal heuristics as supplementary procedures to improve the primal bound during BnB. We turn on the aggressive mode in SCIP to focus on improving the primal bound instead of proving optimality. Details on the implementation of baselines are deferred to Appendix C. To our knowledge, this work is the first ML-guided primal heuristics for general MBQPs.

**Computational Setup** For all methods, we set the time limit to 60s. We report the average PG, PI, and # wins results across 100 test instances for all benchmarks. All models are trained on 800 instances and training is conducted on an NVIDIA A100 GPU with 128 GB of memory. We use the AdamW optimizer (Loshchilov & Hutter, 2017) with learning rate $10^{-5}$ with a batch size of 16. For the combined loss proposed in 3.2, we experiment with $\lambda^{CL} \in \{1, 2, 5, 7\}$ and choose the $\lambda^{CL}$ value that results in the lowest weighted Brier score on the validation set (Details deferred to Appendix G). For all the ML-guided methods, we create sub-MBQPs by fixing the top $p = 0.7$ percent of variables that are least fractional in the predictions at inference time. We also perform a sensitivity analysis of $p \in \{0.65, 0.75\}$ and show that our conclusion holds for most benchmarks for different values of $p$ (Appendix D). Testing (including ML inference and non-ML primal heuristics) is conducted on a cluster with epyc-7542 CPUs with 10 GB RAM. We use SCIP (v8.0.1) (Bestuzheva et al., 2021) for solving the sub-MBQPs.

## 4.2 RESULTS AND DISCUSSION

Table 1: **Data collection statistics with Proposed Randomized Relax-Search vs SCIP.** Average number of distinct solutions found by SCIP ($|\mathcal{S}^+|_{\text{SCIP}}$), average number of distinct solutions in the good samples set $\mathcal{S}^+$ returned by Randomized Relax-Search ($|\mathcal{S}^+|_{\text{Rand-Relax-Search}}$), average objective value found by SCIP (obj. $\mathcal{S}^+_{\text{SCIP}}$), average objective in $\mathcal{S}^+$ by Randomized Relax-Search (obj. $\mathcal{S}^+_{\text{Rand-Relax-Search}}$), and average percentage of variables that take the same value in positive and negative sample pairs (frac. $\mathcal{U}$). The time limit for both strategies is 11000s (details in Appendix G).

| Benchmark | $|\mathcal{S}^+|_{\text{SCIP}}$ | $|\mathcal{S}^+|_{\text{Rand-Relax-Search}}$ | obj. $\mathcal{S}^+_{\text{SCIP}}$ | obj. $\mathcal{S}^+_{\text{Rand-Relax-Search}}$ | frac. $\mathcal{U}$ |
|---|---|---|---|---|---|
| CBQP | 2.01 | 10 | -289048.26 | -887895.29 | 59.71% |
| QMKP | 4.31 | 10 | -46736.29 | -177384.21 | 72.48% |
| CQKP | 2.47 | 10 | -58559.04 | -187403.44 | 56.74% |
| WFLOP | 8.59 | 10 | 1999.23 | 1478.76 | 57.06% |

**Training Data collection** We compare the training data collection results with the Randomized Relax-Search procedure proposed in 3.3 and running the SCIP solver under the same time limit. As shown in Table. 1, the number of distinct solutions found by SCIP throughout the time limit is less than 10 across all benchmarks. Moreover, Randomized Relax-Search produces solutions with better (i.e., lower for minimization problems) objective values in the positive sample set $S^+$. Additionally, we compute the average fraction of the subset of variables that take the same value in positive and negative sample pairs discussed in Proposition 1 (frac. $\mathcal{U} = \frac{1}{N} \sum_{i=1}^{N} \frac{1}{|\mathcal{S}^+_{M_i}|} \sum_{x^+ \in \mathcal{S}^+_{M_i}} \frac{|\mathcal{U}^i_{x^+}|}{|\mathcal{B}_i|}$). The average fraction is greater than $50\%$ in all the four MBQP benchmarks studied, which justifies our proposed extension of combining CL and WCE losses to improve the solution prediction performance for this subset.

**Inference** As shown in Table. 2, both the adapted WCE method and the extend CL+WCE method outperform the baselines in terms of average PG and PI. The adapted CL-based method fails to

produce feasible solutions for a large number of instances in QMKP, CQKP, and WFLOP. The extended CE+WCE method performs the best in terms of PG, PI, and number of wins in PI in all benchmarks studied and significantly improves both feasibility and solution quality compared to the adapted CL-based method. We also show the progress of PG over time in Fig. 2. The proposed CL+WCE method finds better solutions across all time steps in most benchmarks (with the exception that SCIP and RENS have lower PG in the beginning in WFLOP), with the adapted WCE method being the second-best.

Table 2: **Primal Gap (PG), Primal Integral (PI), and # wins in terms of PI.** Lower is better for PG and PI. Higher is better for # wins. WCE and CL are ML-guided MBQP primal heuristics that we adapted from MILPs. CL+WCE is the extended ML method with the proposed combined loss function. $^\dagger$ indicates benchmarks where there are instances for which the method did not produce a feasible solution. For CL, the the feasibility rates are 2%, 0%, and 23% for QMKP, CKQP, and WFLOP. For RENS, the feasibility rates are 65% and 89% for CBQP and CQKP. For all other methods and benchmarks, the feasibility rate is 100%. Note that the PG, PI, and # wins metrics in Section 4.1 apply to all instances regardless of feasibility.

| | | CBQP | | | QMKP | | | CQKP | | | WFLOP | | |
| | Method | PG | PI | # wins | PG | PI | # wins | PG | PI | # wins | PG | PI | # wins |
|---|---|---|---|---|---|---|---|---|---|---|---|---|---|
| Adapted | WCE | 0.15 | 34.13 | 20 | 0.15 | 27.5 | 26 | 0.2 | 37.26 | 12 | 0.37 | 40.16 | 20 |
| | CL | 0.33 | 41.43 | 6 | 0.98$^\dagger$ | 59.22$^\dagger$ | 0$^\dagger$ | 1$^\dagger$ | 60$^\dagger$ | 0$^\dagger$ | 0.71$^\dagger$ | 53.83$^\dagger$ | 5$^\dagger$ |
| Extended | CL+WCE | **0.06** | **23.76** | **74** | **0.11** | **24.24** | **74** | **0.09** | **28.21** | **79** | **0.23** | **36.49** | **37** |
| Baselines | SCIP | 1 | 60 | 0 | 0.9 | 58.04 | 0 | 1 | 60 | 0 | 0.72 | 46.13 | 6 |
| | RENS | 1$^\dagger$ | 60$^\dagger$ | 0$^\dagger$ | 0.99 | 59.94 | 0 | 0.99$^\dagger$ | 59.67$^\dagger$ | 0$^\dagger$ | 0.4 | 42.11 | 22 |
| | Undercover | 1 | 60 | 0 | 1 | 60 | 0 | 1 | 60 | 0 | 0.98 | 59.4 | 0 |
| | Relax-Search | 0.57 | 46.05 | 0 | 0.63 | 51.8 | 0 | 0.5 | 44.07 | 9 | 0.57 | 45.78 | 10 |

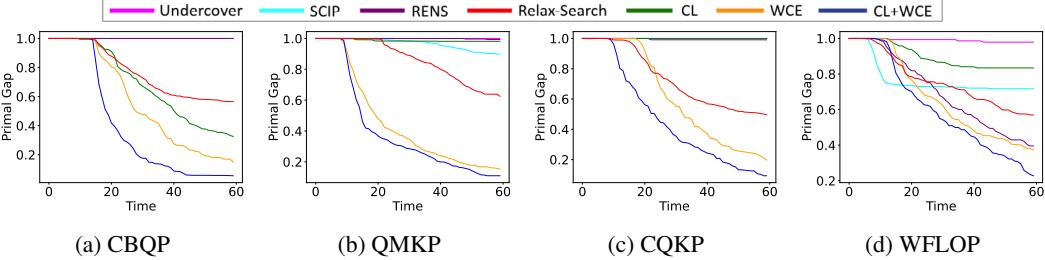

| (a) CBQP | (b) QMKP | (c) CQKP | (d) WFLOP |

Figure 2: Primal gap as a function of time (lower is better).

## 5 CROSS-REGIONAL GENERALIZATION ON WFLOP

The purpose of this study is to understand the impact of regional wind-pattern heterogeneity on the inference performance of the ML-guided primal MBQP heuristics. Wind farm layout optimization in offshore environments is greatly impacted by regional differences in wind resources caused by geographic factors such as proximity to shore, water depth, and local climate. This study attempts to understand the generalization capacity of ML-based primal heuristic methods to unseen regions. The goal is to develop methods that generalize effectively, enabling the efficient design of wind farms using regional wind scenario distributions, even in regions with limited or no historical wind data for validation. To quantify the cross-regional transferability, we develop the ML-guided primal heuristics using training instances across five U.S. regions and test on multiple unseen regions.

**Setup** For the ML-guided solution methods, we train the ML models on a mix of 800 instances randomly selected from five regions: Pacific Northwest, Mid Atlantic, Maine, Gulf of Mexico, and South Atlantic. At inference time, we use the trained models to predict solutions for WFLOP instances from three unseen regions: California, Hawaii, and Great Lakes (100 test instances per region). We use the same instance generation method and computational settings as in Section 4.1.

Table 3: **Cross-regional generalization results on WFLOP. Primal Gap (PG), Primal Integral (PI) results, and # wins in PI.** Lower is better for PG and PI. Higher is better for # wins. $^\dagger$ indicates benchmarks where there are instances for which the method did not produce a feasible solution. For CL, the feasibility rates are 36%, 34%, and 13% for California, Hawaii, and Great Lakes. For Undercover, the feasibility rates are 99%, 0%, and 0% for California, Hawaii, and Great Lakes. For Relax-Search, the feasibility rate is 0% for all 3 regions.

| | | California | | | Hawaii | | | Great Lakes | | |
|---|---|---|---|---|---|---|---|---|---|---|
| | Method | PG | PI | # wins | PG | PI | # wins | PG | PI | # wins |
| Adapted | WCE | 0.33 | 39.55 | 24 | 0.54 | 51.93 | 3 | 0.51 | 51.89 | 6 |
| | CL | 0.62$^\dagger$ | 49.78$^\dagger$ | 13$^\dagger$ | 0.62$^\dagger$ | 50.89$^\dagger$ | 26$^\dagger$ | 0.72$^\dagger$ | 56.18$^\dagger$ | 10$^\dagger$ |
| Extended | CL+WCE | **0.32** | **36.45** | **32** | **0.32** | **43.93** | **34** | **0.32** | **43.71** | **43** |
| Baselines | SCIP | 0.72 | 46.56 | 8 | 0.73 | 46.51 | 15 | 0.81 | 50.79 | 4 |
| | RENS | 0.4 | 40.93 | 23 | 0.55 | 46.35 | 22 | 0.47 | 44.08 | 37 |
| | Undercover | 0.99$^\dagger$ | 59.7$^\dagger$ | 0$^\dagger$ | 1$^\dagger$ | 60$^\dagger$ | 0$^\dagger$ | 1$^\dagger$ | 60$^\dagger$ | 0$^\dagger$ |
| | Relax-Search | 1$^\dagger$ | 60$^\dagger$ | 0$^\dagger$ | 1$^\dagger$ | 60$^\dagger$ | 0$^\dagger$ | 1$^\dagger$ | 60$^\dagger$ | 0$^\dagger$ |

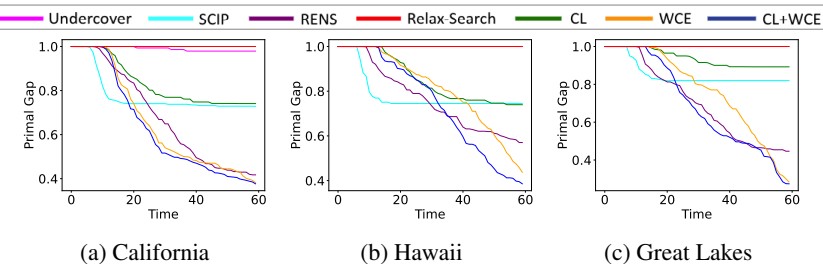

| (a) California | (b) Hawaii | (c) Great Lakes |
|---|---|---|

Figure 3: Primal gap as a function of time (lower is better).

**Results and Discussion.** As shown in Table. 3 and Fig. 3, RENS is a strong baseline for WFLOP. While the ML-based WCE method adapted from MILPs outperform RENS in the in-domain inference in Section 4.2, it fails to outperform RENS on the Hawaii and Great Lakes groups in the cross-regional study. However, our proposed extension that combines CL and WCE losses improves cross-regional generalization and outperform RENS (and all other methods) in terms of PG, PI, and # wins across all three regions. The success of ML-guided methods in making high quality wind farm layout predictions in unseen regions indicates that the distributional summarization of 10 years of simulated wind data can adequately represent relevant attributes of offshore wind patterns across geographically different regions. This may be due to the spatiotemporal dependence between weather phenomena in the underlying wind simulation models used to generate the data. We show an example of final wind farm layouts predicted by the best performing methods in Table 3 in each category in Appendix H.

## 6 CONCLUSION AND DISCUSSION

We develop ML-guided primal heuristics for MBQPs based on solution prediction. We adapt existing methods on ML-guided MILP primal heuristics to MBQPs by introducing a tripartite graph representation for MBQPs, a neural network architecture for feature extraction, and a data collection procedure that produces diverse high-quality solutions for training. In addition, we extend existing loss functions used in CO solution prediction and combine CL and WCE losses. Experimental results show that the adapted and extended ML-guided methods quickly produce high-quality solutions within a short amount of time and show faster convergence compared to non-ML primal heuristics. Furthermore, our extension with combined losses significantly improves the performance compared to other ML methods adapted with existing loss functions and improves generalization in cross-regional inference on WFLOP instances. A potential limitation with our methods is that feasibility is not guaranteed, while it can be adjusted with variable fixing rates in creating the sub-MBQPs at inference time. For future work, we plan to extend our methods to Mixed-Integer Quadratically Constrained Programs, which is a broader class where feasibility is more challenging. A possible extension is to combine solution prediction and projection methods that restore feasibility.

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

## A  MBQP BENCHMARKS

### A.1  FORMULATION

#### A.1.1  CARDINALITY-CONSTRAINED BINARY QUADRATIC PROGRAMS (CBQP) (ZHENG ET AL., 2012)

$$\min -\sum_{i=1,\dots,n}\sum_{j=1,\dots,n} q_{ij}x_i x_j$$

$$\text{s.t.} \sum_{j=1,\dots,n} x_i = k$$

$$x_i \in \{0,1\}, \quad i=1,\dots,n$$

where $n$ is the number of binary variables and $q_{ij}$ are entries in a symmetric matrix.

#### A.1.2  QUADRATIC MULTIDIMENSIONAL KNAPSACK PROBLEM (QMKP) (FORRESTER & HUNT-ISAAK, 2020)

$$\min -\sum_{i=1}^{n}\sum_{j=1}^{n} q_{ij}x_i x_j - \sum_{i=1}^{n} c_i x_i$$

$$\text{s.t.} \sum_{i=1}^{n} a_{ik}x_i \leq b_k, \quad k=1,\dots,m$$

$$x_i \in \{0,1\}, \quad i=1,\dots,n$$

#### A.1.3  CARDINALITY-CONSTRAINED QUADRATIC KNAPSACK PROBLEM (CQKP) (LÉTOCART ET AL., 2014)

$$\min -\sum_{i=1}^{n}\sum_{j=1}^{n} q_{ij}x_i x_j - \sum_{i=1}^{n} c_i x_i$$

$$\text{s.t.} \sum_{i=1}^{n} a_{ik}x_i \leq b_k, \quad k=1,\dots,m$$

$$\sum_{j=1,\dots,n} x_i = k$$

$$x_i \in \{0,1\}, \quad i=1,\dots,n$$

### A.1.4 WIND FARM LAYOUT OPTIMIZATION PROBLEM (WFLOP) (HUANG ET AL., 2025)

Let $J$ be the set of candidate locations for turbine placement and $K$ be the number of turbines to be installed. The binary decision variable $y_j$ takes the value 1 if a wind turbine is installed at location $j$, and 0 otherwise. The set $J$ is a two-dimensional grid with a given resolution that represents the design area of the wind farm. Let $U$ and $\theta$ represent the free stream wind speed and wind direction, respectively. Let $M$ be a set of wind scenarios $\mathcal{M} = \{1, 2, \ldots, M\}$ drawn from a joint probability distribution $p(U, \theta)$. Each scenario $m$ consists of a wind speed $U^{(m)}$ and wind direction $\theta^{(m)}$ with probability $p^{(m)}$ such that $\sum_{m \in \mathcal{M}} p^{(m)} = 1$. The pairwise wind speed deficit interactions $d_{ij}^{(m)}$ are a function of additional parameters, including wind direction $\theta^{(m)}$. WFLOP MBQP minimizes expected wind speed losses.

$$
\begin{aligned}
\min \quad & \sum_{m \in \mathcal{M}} p^{(m)} U^{(m)} \sum_{j \in J} \sum_{i \in J} \left( d_{ij}^{(m)} \right)^2 y_i y_j \\
\text{s.t.} \quad & \sum_{j \in J} y_j = K \\
& y_j \in \{0, 1\} \quad \forall j \in J
\end{aligned}
$$

### A.2 INSTANCE STATISTICS

The instance statistics are shown in Table. 4.

| Category | Benchmark | # Var | # Cons | Quad. den. |
|---|---|---|---|---|
| Synthetic | CBQP | 1000 | 1 | 25.00% |
| Synthetic | QMKP | 1000 | 5 | 25.00% |
| Synthetic | CQKP | 1000 | 2 | 25.00% |
| Real-world | WFLOP | 1000 | 1 | 32.42% |

Table 4: Instance statistics. Number of binary variables (# Var), number of constraints (# Cons), and quadratic term density (Quad. den).

## B PROOF

**Proof 1** *Let $\theta$ be sets of weights learned by an ML model and $p(x|\theta; M_i)$ be the vector of predictions of variables prediction by the model. We use $p(x|\theta; M_i)^{\mathcal{U}_i^{x^+}}$ to denote the prediction vector for the variable subset $\mathcal{U}_i^{x^+}$, (i.e., $[p_j(x|\theta; M_i)]_{\forall j \in \mathcal{U}_i^{x^+}}$). Similarly, we use $p(x|\theta; M_i)^{\mathcal{B}_i \setminus \mathcal{U}_i^{x^+}}$ to denote the prediction vector for the subset $\mathcal{B}_i \setminus \mathcal{U}_i^{x^+}$. We use $\tilde{x}^{\mathcal{U}_i^{x^+}}$ to denote the variable assignments in $\mathcal{U}_i^{x^+}$ in a solution $\tilde{x}$ (i.e., $[\tilde{x}_j]_{\forall j \in \mathcal{U}_i^{x^+}}$).*

*By the property of the dot product operation, $\exp\left(x^+ \cdot p(\theta; M_i) / \tau(x^+ | M_i)\right)$ can be decomposed into*

$$
\exp\left( (x^{+\mathcal{U}_i^{x^+}} \cdot p(\theta; M_i)^{\mathcal{U}_i^{x^+}} + x^{+\mathcal{B}_i \setminus \mathcal{U}_i^{x^+}} \cdot p(\theta; M_i)^{\mathcal{B}_i \setminus \mathcal{U}_i^{x^+}}) / \tau(x^+ | M_i) \right)
$$

$$
= \exp\left( x^{+\mathcal{U}_i^{x^+}} \cdot p(\theta; M_i)^{\mathcal{U}_i^{x^+}} / \tau(x^+ | M_i) \right) \exp\left( x^{+\mathcal{B}_i \setminus \mathcal{U}_i^{x^+}} \cdot p(\theta; M_i)^{\mathcal{B}_i \setminus \mathcal{U}_i^{x^+}} / \tau(x^+ | M_i) \right)
$$

*Similarly, $\exp\left(\tilde{x} \cdot p(\theta; M_i) / \tau(x^+ | M_i)\right)$ can be expressed as*

$$
\exp\left( x^{+\mathcal{U}_i^{x^+}} \cdot p(\theta; M_i)^{\mathcal{U}_i^{x^+}} / \tau(x^+ | M_i) \right) \exp\left( \tilde{x}^{\mathcal{B}_i \setminus \mathcal{U}_i^{x^+}} \cdot p(\theta; M_i)^{\mathcal{B}_i \setminus \mathcal{U}_i^{x^+}} / \tau(x^+ | M_i) \right)
$$

*. According to Eqn (5).*

$$
\mathcal{L}^+\left(\theta \mid x^+, M_i\right) = -\log \frac{\exp\left( x^{+\mathcal{B}_i \setminus \mathcal{U}_i^{x^+}} \cdot p(\theta; M_i)^{\mathcal{B}_i \setminus \mathcal{U}_i^{x^+}} / \tau(x^+ | M_i) \right)}{\sum_{\tilde{x} \in S_{M_i}^- \cup \{x^+\}} \exp\left( \tilde{x}^{\mathcal{B}_i \setminus \mathcal{U}_i^{x^+}} \cdot p(\theta; M_i)^{\mathcal{B}_i \setminus \mathcal{U}_i^{x^+}} / \tau(x^+ | M_i) \right)}. \tag{7}
$$

Therefore, $\mathcal{L}^{+}\left(\theta \mid x^{+}, M_i\right)$ is a function of $p(\theta; M_i)^{\mathcal{B}_i \setminus \mathcal{U}_i^{x^+}}$. $\qquad\square$

## C BASELINES

For the SCIP solver baseline, we experimented with both Linear Programming (LP) reformulation (default) and Nonlinear Programming (NLP) formulation. We report the results with the default settings, as we found that the performance with the default LP reformulation is stronger than the NLP formulation. For Relax-Search (Huang et al., 2025), we experimented with both LP and NLP relaxation as the basis. We report the results with LP relaxation as the basis, as the performance of LP relaxation is stronger in our experiments. For the Undercover and RENS baselines, we use the official implementation of RENS and Undercover in SCIP. To run standalone RENS and Undercover in SCIP, we apply RENS/Undercover at the root node and disable all other primal heuristics.

## D SENSITIVITY ANALYSIS OF PROPOSED METHODS

We perform a sensitivity analysis of $p \in \{0.65, 0.75\}$, as shown in Table 5 and Table 6.

Table 5: Primal Gap (PG), Primal Integral (PI), and # wins in terms of PI with $p = 0.65$. [†] indicates benchmarks where there are instances for which the method did not produce a feasible solution. For CL, the the feasibility rates are 2%, 0%, and 24% for QMKP, CKQP, and WFLOP. For RENS, the feasibility rates are 65% and 89% for CBQP and CQKP. For all other methods and benchmarks, the feasibility rate is 100%.

| | | CBQP | | | QMKP | | | CQKP | | | WFLOP | | |
|---|---|---|---|---|---|---|---|---|---|---|---|---|---|
| | Method | PG | PI | # wins | PG | PI | # wins | PG | PI | # wins | PG | PI | # wins |
| Adapted | WCE | 0.42 | 47.81 | 11 | 0.23 | 35.46 | 38 | 0.56 | 52.42 | 9 | 0.55 | 48.25 | 3 |
| | CL | 0.34 | 40.82 | 13 | 0.98[†] | 59.31[†] | 0[†] | 1[†] | 60[†] | 0[†] | 0.7[†] | 53.4[†] | 10[†] |
| Extended | CL+WCE | 0.09 | 31.69 | 71 | 0.28 | 32.97 | 60 | 0.31 | 45.18 | 51 | 0.45 | 44.67 | 25 |
| Baselines | SCIP | 1 | 60 | 0 | 0.9 | 58.04 | 0 | 1 | 60 | 0 | 0.72 | 46.13 | 13 |
| | RENS | 1[†] | 60[†] | 0[†] | 0.99 | 59.94 | 0 | 0.99[†] | 59.67[†] | 0[†] | 0.4 | 42.11 | 34 |
| | Undercover | 1 | 60 | 0 | 1 | 60 | 0 | 1 | 60 | 0 | 0.98 | 59.4 | 0 |
| | Relax-Search | 0.57 | 46.05 | 5 | 0.63 | 51.8 | 2 | 0.5 | 44.07 | 40 | 0.57 | 45.78 | 16 |

Table 6: Primal Gap (PG), Primal Integral (PI), and # wins in terms of PI with $p = 0.75$. [†] indicates benchmarks where there are instances for which the method did not produce a feasible solution. For CL, the the feasibility rates are 2%, 0%, and 18% for QMKP, CKQP, and WFLOP. For CL+WCE, the feasibility rate is 92% for QMKP. For RENS, the feasibility rates are 65% and 89% for CBQP and CQKP. For all other methods and benchmarks, the feasibility rate is 100%.

| | | CBQP | | | QMKP | | | CQKP | | | WFLOP | | |
|---|---|---|---|---|---|---|---|---|---|---|---|---|---|
| | Method | PG | PI | # wins | PG | PI | # wins | PG | PI | # wins | PG | PI | # wins |
| Adapted | WCE | 0.05 | 25.81 | 26 | 0.16 | 24.61 | 41 | 0.18 | 30.24 | 18 | 0.45 | 40.78 | 21 |
| | CL | 0.33 | 32.15 | 0 | 0.98[†] | 59.19[†] | 0[†] | 1[†] | 60[†] | [†]0 | 0.74[†] | 55.15[†] | 3[†] |
| Extended | CL+WCE | 0.03 | 19.35 | 74 | 0.19[†] | 25.87[†] | 59[†] | 0.1 | 25.79 | 79 | 0.31 | 33.31 | 41 |
| Baselines | SCIP | 1 | 60 | 0 | 0.9 | 58.04 | 0 | 1 | 60 | 0 | 0.72 | 46.13 | 6 |
| | RENS | 1[†] | 60[†] | 0[†] | 0.99 | 59.94 | 0 | 0.99[†] | 59.67[†] | 0[†] | 0.4 | 42.11 | 19 |
| | Undercover | 1 | 60 | 0 | 1 | 60 | 0 | 1 | 60 | 0 | 0.98 | 59.4 | 0 |
| | Relax-Search | 0.57 | 46.05 | 5 | 0.63 | 51.8 | 2 | 0.5 | 44.07 | 40 | 0.57 | 45.78 | 16 |

Additionally, we report the results with $p = 1.0$, which shows the performance using the full predictions from the ML models, without and refinement (Table 7). Feasibility rates for the ML methods are shown in Table. 8. Note that the sum of # wins across all methods can be larger than 100 (the total number of test instances) when there are ties.

Table 7: Primal Gap (PG), Primal Integral (PI), and # wins in terms of PI with $p = 1.0$. $\dagger$ indicates benchmarks where there are instances for which the method did not produce a feasible solution. For RENS, the feasibility rates are 65% and 89% for CBQP and CQKP. Feasibility rates for the ML methods are shown in Table. 8. For all other methods and benchmarks, the feasibility rate is 100%.

| | | CBQP | | | QMKP | | | CQKP | | | WFLOP | | |
| | Method | PG | PI | # wins | PG | PI | # wins | PG | PI | # wins | PG | PI | # wins |
|---|---|---|---|---|---|---|---|---|---|---|---|---|---|
| Adapted | WCE | $1^\dagger$ | $59.83^\dagger$ | $30^\dagger$ | $0.77^\dagger$ | $48.75^\dagger$ | $46^\dagger$ | $1^\dagger$ | $60^\dagger$ | $1^\dagger$ | $0.54^\dagger$ | $60^\dagger$ | $0^\dagger$ |
| | CL | $1^\dagger$ | $60^\dagger$ | $28^\dagger$ | $1^\dagger$ | $59.88^\dagger$ | $0^\dagger$ | $1^\dagger$ | $60^\dagger$ | $1^\dagger$ | $0.54^\dagger$ | $60^\dagger$ | $0^\dagger$ |
| Extended | CL+WCE | $1^\dagger$ | $59.81^\dagger$ | $28^\dagger$ | $1^\dagger$ | $60^\dagger$ | $0^\dagger$ | $1^\dagger$ | $60^\dagger$ | $1^\dagger$ | $0.54^\dagger$ | $60^\dagger$ | $0^\dagger$ |
| Baselines | SCIP | 1 | 60 | 28 | 0.9 | 58.04 | 1 | 1 | 60 | 1 | 0.72 | 46.13 | 31 |
| | RENS | $1^\dagger$ | $60^\dagger$ | $28^\dagger$ | 0.99 | 59.94 | 0 | $0.99^\dagger$ | $59.67^\dagger$ | $1^\dagger$ | 0.4 | 42.11 | 52 |
| | Undercover | 1 | 60 | 28 | 1 | 60 | 0 | 1 | 60 | 1 | 0.98 | 59.4 | 0 |
| | Relax-Search | 0.57 | 46.05 | 71 | 0.63 | 51.8 | 53 | 0.5 | 44.07 | 100 | 0.57 | 45.78 | 20 |

Table 8: Feasibility rates for the adapted and extended ML methods with $p = 1.0$.

| | CBQP | QMKP | CQKP | WFLOP |
|---|---|---|---|---|
| WCE | 3% | 98% | 0 | 0 |
| CL | 0 | 1% | 0 | 0 |
| CL+WCE | 3% | 0 | 0 | 0 |

## E PG PLOTS WITH MEDIAN AND QUANTILES

We provide the plots of median PG values as a function of time along with the 25th and 75th percentiles in Fig. 4 and Fig. 5.

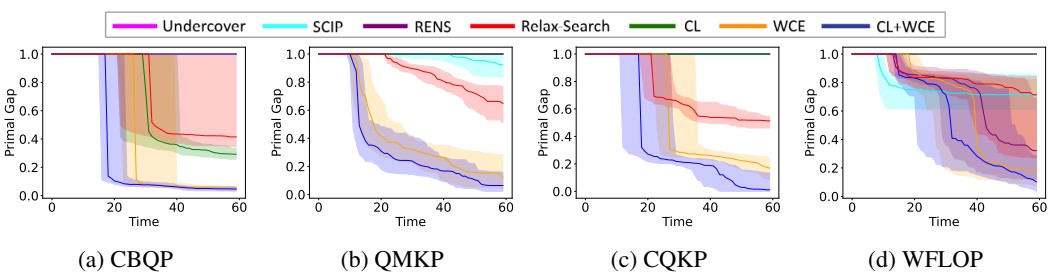

(a) CBQP      (b) QMKP      (c) CQKP      (d) WFLOP

Figure 4: Primal gap as a function of time (lower is better) on four synthetic benchmarks.

## F NEURAL NETWORK ARCHITECTURE

**List of features** The full list of features for the tripartite graph is shown in Table 9.

**GAT module details** For the embedding layers, we use 2-layer MLPs with 64 hidden units per layer and ReLU as the activation function to map the node and edge features $(C, E, V, E', Q)$ to new embeddings $(C_1, E_1, V_1, E'_1, Q_1)$ in $\mathbb{R}^d$ where $d = 64$. The GAT performs four rounds of message passing, as shown in Fig. 1 (C). In round one, each quadratic term node in $Q_1$ attends over its neighbors in $V_1$ using $H$ attention heads to produce updated quadratic term embeddings $Q_2$. In round two, each variable node in $V_1$ attends over its neighbors (using a separate set of $H$ heads) to produce updated variable embeddings $V_2$. In round three, each constraint node in $C_1$ attends over its neighbors in $V_2$ to produce updated constraint embeddings $C_2$. In the final round, each variable node in $V_2$ attends over its neighbors in $C_2$ to produce the final variable embeddings $V_3$. We use $H = 8$ attention heads.

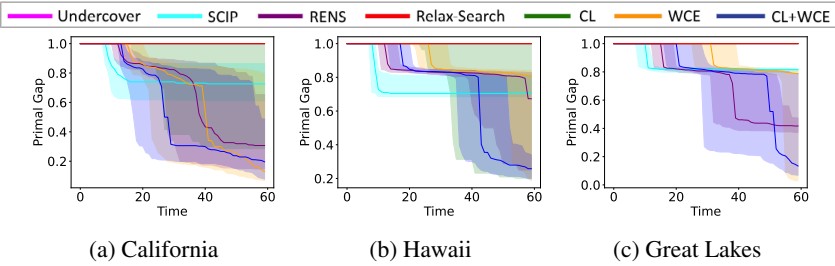

(a) California         (b) Hawaii         (c) Great Lakes

Figure 5: Primal gap as a function of time (lower is better) in cross-regional WFLOP transferability experiment benchmarks.

| Nodes | Features | Source |
|---|---|---|
| C | avg. coefficients in the constraint | (Han et al., 2023) |
| | min. coefficients in the constraint | new |
| | max. coefficients in the constraint | new |
| | variance of coefficients in the constraint | new |
| | # of variables in the constraint | (Han et al., 2023) |
| | left-hand side or right-hand side | (Han et al., 2023) |
| | constraint sense in one-hot encoding (3) $(=, >, <)$ | new |
| V-C edge | coefficient of variables in constraints | (Han et al., 2023) |
| V | normalized coefficient in obj (among linear terms) | (Han et al., 2023) |
| | avg. coefficient in constraints | (Han et al., 2023) |
| | # of times it appear in linear constraints | (Han et al., 2023) |
| | variance of. coefficient in constraints | new |
| | max. coefficient in constraints | (Han et al., 2023) |
| | min. coefficient in constraints | (Han et al., 2023) |
| | binary variable indicator | (Han et al., 2023) |
| | LP relaxation value in MILP reformulation | new |
| | # times it appears in quadratic terms | new |
| | avg. coefficient in quadratic terms that it appears in | new |
| | max. coefficient in quadratic terms that it appears in | new |
| | min. coefficient in quadratic terms that it appears in | new |
| | variance of coefficient in quadratic terms that it appears in | new |
| | avg. # times its neighbors appears in quadratic terms | new |
| | max. # times its neighbors appears in quadratic terms | new |
| | min. # times its neighbors appears in quadratic terms | new |
| | variance of # times its neighbors appears in quadratic terms | new |
| | Eigenvalue centrality in Hessian graph | new |
| Q | coefficient of quadratic term in objective function | new |
| | LP relaxation value of reformulated variable $z_{ij} = x_i x_j$ | new |
| | LP relaxation violation | new |
| | Edge centrality in Hessian graph | new |
| V-Q edge | None | new |

Table 9: Features of MBQP tripartite graph representation.

# G   DATA COLLECTION AND TRAINING

## G.1   DATA COLLECTION

For data collection, we set a Nonlinear Programming relaxation time limit of $T_r = 1000s$, a subproblem time limit of $T_s = 1000s$, number of random seeds $K = 10$, candidate fixing ratio of $p_1 = 0.9$, and final fixing ratio of $p_2 = 0.7$. The total time limit of data collection for each instance is 11000s, as use $T_r = 1000s$ and $T_s = 1000s$ with 10 random seeds. We solve the NLP relaxation problems using epyc-9354 CPUs with 300 GB RAM and the MBQP subproblems on epyc-7542 CPUs with 10 GB RAM.

## G.2 TRAINING

For each MBQP benchmark, 800 instances are used for training and 100 are used for validation. For the cross-regional WFLOP transferability experiment in Section 5, the training set contains 800 instances uniformly randomly selected from 5 regions: Pacific Northwest, Mid Atlantic, Maine, Gulf of Mexico, and South Atlantic, and the validation set contains 100 instances randomly selected from the same five training regions. We train the models for 2000 epochs.

## G.3 CL WEIGHT HYPERPARAMETERS

**Weighted Brier score** We use a weighted Brier score where the Brier score with respect to each positive sample $x^+ \in S_{M_i}^+$ is weighted by the energy function in Eqn. 2. Formally, the weighted Brier score is

$$BS = \frac{1}{N} \sum_{t=1}^{N} \sum_{x^+ \in S_{M_i}^+} \sum_{d \in B_i} w(x^+|M_i)(p_d(x|\theta, M_i) - t_i^+)^2$$

where $t_{i,d}^+$ is the solution assignment of variable indexed by $d$ in sample $x^+$ in $M_i$. $B_i$ is the set of binary variables in $M_i$. $N$ is the total number of instances in the validation set. For the proposed CL+WCE method, we experiment with $\lambda^{CL} \in \{1, 2, 5, 7\}$ and choose the $\lambda^{CL}$ value that results in the lowest weighted Brier score on the validation set. The value of $\lambda^{CL}$ is 1, 2, 5, and 2 for CBQP, QMKP, CQKP, and WFLOP-California. $\lambda^{CL}$ is 7 for WFLOP-mix.

## H EXAMPLE TURBINE LAYOUT SOLUTION IN CROSS-REGIONAL WFLOP TRANSFERABILITY EXPERIMENTS

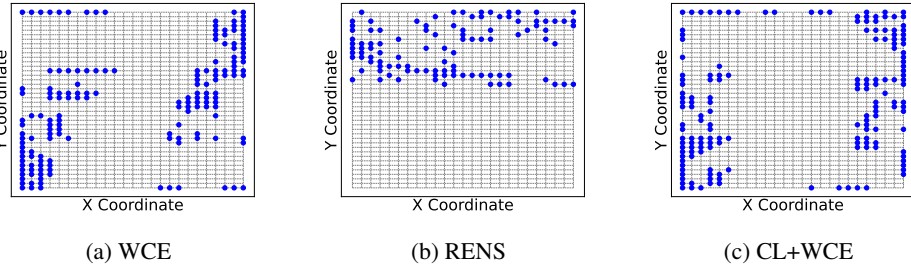

|          (a) WCE          |          (b) RENS          |          (c) CL+WCE          |

Figure 6: Final wind farm layout solutions for a location in the Hawaii region predicted by different algorithms. Filled circles represent a wind turbine at that grid location.

In Figure 6 we show example final wind farm layouts predicted by the best performing methods in Table 3 in each category: adapted ML (WCE), extended ML (CL+WCE), and non-ML (RENS). The CL+WCE layout solution leads to a 17% reduction in wind speed losses due to wake effects compared to the WCE layout solution (the second best solution). Wind speed losses impact power production and are a function of turbine placement. And although it is also a function on the distribution of the wind scenarios, greater turbine spacing will often lead to lessened wake effects and less wind speed loss. Thus, the improvement in objective value in the design in Fig. 6(c) may be attributed to the slightly more dispersed turbine clustering.

