# OpenReview forum: "ML-Guided Primal Heuristics for Mixed Binary Quadratic Programs"
_ICLR.cc/2026/Conference — Submitted to ICLR 2026_

### Official Review · Reviewer_n1H8 · 2025-10-24

**Soundness:** 3
**Presentation:** 3
**Contribution:** 2
**Rating:** 4
**Confidence:** 4

**Summary:**

The paper proposes ML-guided primal heuristics for mixed binary quadratic programs by encoding each instance as a tripartite graph  and using Graph attention to predict assignments to binary variables. Training combines cross-entropy and contrastive losses with a Randomized Relax-Search data generator; inference performs Neural Diving by fixing high-confidence variables before solving the reduced problem.

**Strengths:**

1. The Randomized Relax-Search framework for solution generation is novel and practical. It could enable efficient sampling of large, diverse candidate sets for challenging MBQPs and yielding richer supervision signals.
2. The empirical performance against baselines is relatively strong, indicating that the proposed pipeline translates into effective primal heuristics.

**Weaknesses:**

Although the performance of the proposed framework is promising, the significance of this work remains questionable from the following aspects:
1. Excluding the data-generation component, the method is more like a straightforward combination of [1], [2], and [3]. It would be nice to see the author explaining about what new capability or insight emerges from this integration beyond the sum of parts?
2. Since tripartite QP encodings is not new (e.g., hypernode constructions in [4]), a short comparison would help: what is distinct in the author's design, and how do those choices benefit MBQPs?
3. Baselines are limited to conventional algorithms. Methods like [1] can be adapted to MBQPs given sufficient data. Is it possible to include such learning-based baselines or justify their exclusion with a discussion of expected performance.

[1] Nair, Vinod, et al. "Solving mixed integer programs using neural networks." arXiv preprint arXiv:2012.13349 (2020).
[2] Han, Qingyu, et al. "A gnn-guided predict-and-search framework for mixed-integer linear programming." arXiv preprint arXiv:2302.05636 (2023).
[3] Huang, Taoan, et al. "Contrastive predict-and-search for mixed integer linear programs." Forty-first International Conference on Machine Learning. 2024.
[4] Wu, Chenyang, et al. "On representing convex quadratically constrained quadratic programs via graph neural networks." arXiv preprint arXiv:2411.13805 (2024).

**Questions:**

Besides of those mentioned in the weakness section, there are some other questions.
1. Why are commercial solvers (e.g., Gurobi) excluded from the baseline? SCIP is often weaker on nonconvex MIQPs; including Gurobi (and others) would provide a more balanced baseline set.
2. How many solutions were collected by the Randomized Relax-Search procedure? Is it possible to report the total counts to contextualize the dataset sizes and training signal diversity.

---

### Official Review · Reviewer_LE28 · 2025-10-30

**Soundness:** 3
**Presentation:** 3
**Contribution:** 2
**Rating:** 4
**Confidence:** 4

**Summary:**

This work applies a combination of established Learning-to-Optimize (L2O) techniques to the MIQP domain. While the integration is competently executed, the paper would benefit from a clearer articulation of its core novel contribution, as the current approach appears to be a straightforward application of existing concepts rather than a significant algorithmic innovation.

**Strengths:**

This paper is fairly easy to follow, it combines the state-of-the-art L2O techniques.

**Weaknesses:**

- While the paper competently integrates several existing techniques—such as graph representations for optimization problems, distribution learning, and contrastive learning—the core novelty of the overall framework appears limited. The application seems to be a straightforward combination of these methods without significant customization for the MIQP structure itself. This concern is compounded by the experimental design: the absence of a state-of-the-art commercial solver like Gurobi as a baseline makes it difficult to assess the method's practical utility.

- Furthermore, benchmarking on a more challenging and diverse dataset, such as QPLIB [1], would provide a more rigorous validation of the approach's robustness and generalizability.

[1] https://qplib.zib.de

**Questions:**

- Regarding the graph representation, did the authors consider the alternative of creating direct edges between variable nodes instead of introducing quadratic nodes? This simpler representation, which uses significantly fewer nodes, has been successfully applied to Mixed-Integer Quadratic Programs (MIQPs) in prior work [1].

[1] Chen, Ziang, Xiaohan Chen, Jialin Liu, Xinshang Wang, and Wotao Yin. "Expressive Power of Graph Neural Networks for (Mixed-Integer) Quadratic Programs." In Forty-second International Conference on Machine Learning.

---

### Official Review · Reviewer_g6qN · 2025-10-31

**Soundness:** 2
**Presentation:** 3
**Contribution:** 2
**Rating:** 4
**Confidence:** 4

**Summary:**

This paper develops ML-guided primal heuristics for MBQPs by representing an instance as a tripartite graph (constraints, variables, quadratic-term nodes), processing it with a four-round GAT to predict a binary assignment, and solving a reduced sub-MBQP via Neural Diving at inference. Besides, it introduces Randomized Relax-Search to collect diverse positive/negative solutions for training, and combines CL with a CE/WCE term (CL+WCE).

In my opinion, two biggest concerns with L2O methods remain: (1) the expensive training cost for supervised learning, and (2) the lack of feasibility guarantees. In this paper, I do not see convincing evidence that either issue is meaningfully discussed. These gaps materially limit the practical impact of the approach.

**Strengths:**

The tripartite representation and GAT pipeline explicitly model quadratic couplings through Q–V edges and C–V interactions, which is a sensible structural upgrade over MILP-only predictors. The theoretical observation behind Proposition 1 motivates a combined CL+WCE loss that empirically dominates pure CL/WCE across benchmarks and improves feasibility and solution quality in short time budgets. The randomized relax-search procedure yields more and better positive samples than simply logging solver trajectories, strengthening the supervised signal.

**Weaknesses:**

1. The experimental deck emphasizes SCIP and classical heuristics under a 60s limit, but it does not benchmark against tuned commercial MBQP solvers, e.g., CPLEX or Gurobi, or MIQP linearization pipelines, lacking fairness.
2. The training data collection is computationally expensive (11,000s per instance), and the paper does not present a cost–benefit accounting that includes training and maintenance when compared at matched accuracy/latency.
3. Feasibility is not guaranteed and controlled by the fixing ratio 𝑝. Although sensitivity is reported, failure modes for dense/near-infeasible cases are not charted.
4. Finally, scalability beyond n=1000 and different quadratic densities/condition numbers lacks a systematic Pareto analysis.

**Questions:**

1. Can you provide an ablation showing how often the model’s fixed assignments are infeasible before refinement, and how much each mechanism reduces infeasibility across densities and near-infeasible regimes? Also, if the above ratio is very high, what would you do to improve it?
2. Any ideas to address the problem of expensive training data costs? BTW, given such high data and training costs, is supervised learning truly a suitable mode for L2O, especially when dealing with practical large-scale industrial problems?

---

### Official Review · Reviewer_urQJ · 2025-11-02

**Soundness:** 3
**Presentation:** 2
**Contribution:** 2
**Rating:** 6
**Confidence:** 2

**Summary:**

This work proposes an ML-guided primal heuristic for Mixed Binary Quadratic Programs (MBQPs), based on solution prediction and extended from ML-guided methods for Mixed-Integer Linear Programs (MILPs). The network takes an MBQP as input, represented as a tripartite graph, and outputs predicted solutions for the binary decision variables. Compared to MILPs, MBQPs face the combined challenges of combinatorial complexity and nonlinearities. To address these, the authors:
1. propose a neural network design that includes a tripartite graph representation of MBQP instances, a custom feature set, and a graph attention network architecture;
2. introduce a loss function that combines weighted cross-entropy with a contrastive learning term;
3. develop a data collection procedure, Randomized Relax-Search, which generates high-quality solutions as ground-truth training data for large-scale MBQPs.

**Strengths:**

Overall, the writing is clear, particularly the introduction of the proposed methodology.

**Weaknesses:**

The formatting needs improvement. For example, some citations (e.g., “RENS Berthold (2014), Undercover Berthold & Gleixner (2014), and Relax-Search Huang et al. (2025)” in Section 2.1) are missing brackets, and some citations (e.g., “Nair et al. (Nair et al., 2020) and Han et al. (Han et al., 2023)” in Section 2.2) are redundant.

**Questions:**

1. The authors state that the adapted CL-based method performs worse in Table 2 because it fails to produce feasible solutions for many instances. However, why does this method fail to produce feasible solutions for QMKP, CQKP, and WFLOP but perform well on CBQP? Why do the adapted WCE and extended CE+WCE methods can produce feasible solutions?
2. The experimental comparison with other ML-guided methods is lacking. Are there any ML-guided methods for MBQPs, or even specifically for the chosen problems?

---

### Author Response · Authors · 2025-12-03
**Extended results with commercial solver and ML baseline**

## Commercial solver (Gurobi)
We were not able to use commercial solvers by the time we submitted the paper because this work was developed during the arthur’s internship at a company where there was no commercial license for commercial solvers such as Gurobi and CPLEX. Only the SCIP solver, which is open-source, was available. Therefore, we used SCIP for data collection (Section 3.3) and evaluation (Section 4.2) in our submission.

However, we have recently open-sourced the code after obtaining permission from the internship company and thus gained access to Gurobi. While we do not have enough time to collect ground truth data using Gurobi and re-train the models with Gurobi data during the rebuttal phase, we added evaluation/inference results with Gurobi. Firstly, we added Gurobi as a separate non-ML baseline (denoted as **Gurobi**). Second, we took the best ML model (CL+WCE) that we used in the submission (which was trained on data collected with SCIP because we did not have access to SCIP by the time we submitted this paper) and solved the sub-MBQPs using Gurobi as a solver (Fig. 1(F)). This method is denoted as **CL+WCE_Gurobi**.

## ML baseline (Surco)

We added Surco (Ferber, Aaron M., et al. 2023), a self-supervised ML-guided primal heuristics for nonlinear combinatorial optimization problems, as a baseline. Surco learns a linear surrogate cost function for nonlinear problems and finds solutions by solving an MILP, using the learned linear surrogate cost. As Surco does not provide a universal module for input feature representation, we use the tripartite graph representation proposed in Section 3.1 in our work to represent input MBQPs and to learn the surrogate linear objective in our implementation.

[1] Ferber, Aaron M., et al. "Surco: Learning linear surrogates for combinatorial nonlinear optimization problems." International Conference on Machine Learning. PMLR, 2023.

## Results
This table below shows the Primal Integral (PI) results for the 4 benchmarks in the experiments in Section 4. Additionally, we compute the average PI across different benchmarks for each method. Note that the PI numbers in the table below are different from the ones in the original manuscript because the PG is normalized by the best objective value found across all methods (which is now updated by CL+WCE_Gurobi and Gurobi).

| Benmark       | CBQP  | QMKP  | CQKP | WFLOP | Average   |
|---------------|-------|-------|-------|-------|-------|
| WCE           |  36.3 |  33.8 | 41.01 | 40.23 | 37.84 |
| CL            | 42.95 |  59.3 |    60 | 55.07 | 54.33 |
| CL+WCE        | 26.81 | 31.06 | 33.41 | 37.43 | 32.18 |
| CL+WCE_Gurobi | **11.91** | **16.95** | **16.77** | 22.63 | **17.07** |
| Surco         | 59.97 |    60 |  58.7 | **20.06** | 49.68 |
| SCIP          |    60 |  58.5 |    60 | 42.55 | 55.26 |
| RENS          |    60 | 59.96 | 59.67 | 40.84 | 55.12 |
| Undercover    |    60 |    60 |    60 | 59.65 | 59.91 |
| Relax-Search  | 47.19 | 53.47 | 46.68 | 42.48 | 47.46 |
| Gurobi        | 27.85 | 23.34 | 29.06 | 55.22 | 33.87 |

CL+WCE_Gurobi performs the best across all the methods in 3 out of the 4 benchmarks and has the lowest average PI across different benchmarks. On the WFLOP benchmark, CL+WCE_GRB performs second-best, with the PI being slightly worse than Surco.

**CL+WCE_Gurobi vs Gurobi.** CL+WCE_Gurobi outperforms Gurobi, even though the ML model in CL+WCE_GRB was trained on ground truth data collected using SCIP (which is considered to be weaker solver than Gurobi). This shows the effectiveness of our ML-guided method. We expect that the performance of CL+WCE_Gurobi will be further improved if the ML model is trained using ground truth data collected with Gurobi. While we are not able to use Gurobi to collect ground truth data and re-train the ML models using Gurobi during this rebuttal phase due to time constraints, we will include the complete results in the final version of the paper if it gets accepted.

**CL+WCE_Gurobi vs Surco.** CL+WCE_Gurobi outperforms Surco in 3 of the 4 benchmarks. CL+WCE_Gurobi reduces the PI by 65.64% compared to Surco on average across the 4 benchmarks, showing the effectiveness of our approach compared to other ML baselines.

---

> ### Author Response · Authors · 2025-12-03
> **Extended results with commercial solver and ML baseline (continued)**
>
> This table below shows the Primal Integral (PI) results for the three regions in the cross-regional WFLOP generalization experiments in Section 5.
>
> | Region        | California | Hawaii | Great Lakes | Average |
> |---------------|------------|--------|-------------|---------|
> | WCE           |      39.45 |  48.16 |        43.4 |   43.67 |
> | CL            |      51.52 |  52.07 |       57.06 |   53.55 |
> | CL+WCE        |      38.91 |  44.99 |       40.93 |   41.61 |
> | CL+WCE_Gurobi |      23.53 |  **17.13** |       23.92 |   **21.53** |
> | Surco         |      **20.36** |   28.1 |        **20.3** |   22.92 |
> | SCIP          |      42.75 |  48.36 |       43.33 |   44.81 |
> | RENS          |      40.73 |  45.48 |       41.36 |   42.52 |
> | Undercover    |      59.56 |     60 |          60 |   59.85 |
> | Relax-Search  |         60 |     60 |          60 |   60.00 |
> | Gurobi        |      56.16 |  58.56 |       57.56 |   57.43 |
>
> In terms of the average PI across three sites, the performance of CL+WCE_Gurobi is stronger than the Surco baseline. CL+WCE_Gurobi performs slightly worse than Surco for the California and Great Lakes regions but significantly outperforms Surco (reducing the PI by 39.04%) on the Hawaii region. CL+WCE_Gurobi also significantly reduces the PI compared to all other methods.

---

### Author Response · Authors · 2025-12-03
**Graph representation of MBQPs**

**Comparison with (Wu, Chenyang, et al. 2024)**

Reviewer n1H8 asked about comparison with the tripartite graph representation in (Wu, Chenyang, et al. 2024).

While both (Wu, Chenyang, et al. 2024) and our work use tripartite graph representations, the types of optimization problems we study are different. Wu, Chenyang, et al. (2024) study Quadratically Constrained Quadratic Programs (QCQPs), which are continuous optimization problems with quadratic terms in both the objective function and constraints. We study MBQPs, which are discrete optimization problems with quadratic terms in the objective function and linear constraints.

Our feature sets for the nodes in the tripartite graph in our design are distinct compared to (Wu, Chenyang, et al. 2024).  MBQPs contain binary decision variables and thus require features that capture the nature of discrete problems. We proposed novel features for MBQPs that capture both the characteristics of the discrete decision variables and quadratic terms, such as LP relaxation violation and binary variable indicator. The full set of features that we used can be found in Table 9 in Appendix F.

The graph structure (without comparing the feature sets for the nodes) in our proposed representation is equivalent to the one proposed in (Wu, Chenyang, et al. 2024) in the special case where there is no quadratic term in the constraints in QCQPs.

**Comparison with (Chen, Ziang, et al. 2024)**

Reviewer LE28 asked about the alternative option of using the graph representation proposed in (Chen, Ziang, et al. 2024).

We did not use the representation proposed in (Chen, Ziang, et al. 2024) for this submission because the representation of creating direct edges between variable nodes can not universally represent general QPs with discrete decision variables in theory, according to (Chen, Ziang, et al. 2024). However, studying the empirical performance of the representation proposed in (Chen, Ziang, et al. 2024) could be an ablation study. We plan to include a comparison with the architecture proposed in (Chen, Ziang, et al. 2024) for the final version of this paper if it gets accepted.

[1] Wu, Chenyang, et al. "On representing convex quadratically constrained quadratic programs via graph neural networks." arXiv preprint arXiv:2411.13805 (2024).

[2] Chen, Ziang, et al. "Expressive power of graph neural networks for (mixed-integer) quadratic programs." arXiv preprint arXiv:2406.05938 (2024).

---

### Author Response · Authors · 2025-12-03

**Formatting**

We have fixed the formatting issues with the citations and updated the manuscript.

**Linear reformulation**

Reviewer g6qN has raised questions about comparison with MIQP linear reformulation, in addition to commercial solvers. SCIP linearizes the product of binary decision variables by default. Therefore, the SCIP baseline that we showed in Section 4 uses linear reformulation. We also experimented with the original nonlinear formulation by disabling the reformulation option in SCIP. We did not include results with the nonlinear formulation in the manuscript because SCIP with the original nonlinear formulation is weaker compared to SCIP with linearization (the default option).

**Number of solutions collected by Randomized Relax-Search**

Reviewer n1H8 asked about the number of solutions collected by Randomized Relax-Search. As we use 10 random seeds (resulting in 10 subproblems) and take the best solution from each subproblem, the number of solutions returned by Randomized-Relax-Search is 10 for each instance. The comparison with the number of solutions returned by SCIP under the same time limit is included in Table 1 in the manuscript. As shown in Table 1, Randomized Relax-Search produces a larger number of diverse solutions and improves the average solution quality for the training data compared to SCIP.

**Training data collection cost**

We acknowledge that the training data collection is computationally expensive in supervised learning. The rationale behind using supervised learning in ML-for-COs is that we can afford to spend a larger amount of time in data collection because data is collected offline and does not cause overhead to the solving times during inference.

Prior work in existing literature has also used self-supervised learning to reduce the cost of data collection. For example, Surco (Ferber, Aaron M., et al. 2023) is a self-supervised ML method that learns a surrogate linear cost function for nonlinear problems. We added a comparison between our methods and Surco (Official Comment titled Extended results with commercial solver and ML baseline). We showed that our approach significantly outperforms Surco in terms of primal integral on average across 4 benchmarks (winning in 3 of the 4 benchmarks). This shows the value of supervised learning, also it comes with a higher training data collection cost.

**Number of binary variables and quadratic term density**

The real-world WFLOP benchmark has an average quadratic density of 32.42%, which is larger than that of the synthetic benchmarks (25%). Our proposed method applies to instances with different numbers of binary variables and quadratic term density. We plan to include experiments with larger instances for the final version of the paper if it gets accepted.

---

### Author Response · Authors · 2025-12-03
**Contribution and innovation**

We believe that we have made substantial contributions in developing customized ML-guided primal heuristics for MBQPs. (1) We designed a novel data collection procedure for MBQPs, which has shown to significantly improve solution quality of training samples and increase the diversity of solutions compared to simply running the solver for a large amount of time (Table 1). (2) We proposed a custom tripartite graph representation for MBQPs that captures the characteristics of both the nonlinear terms and the discrete nature of this problem class. We also developed a convolution mechanism with four rounds of message passing for input feature extraction for this tripartite graph representation (Appendix F).  (3) We provided a theoretical observation for the reason why CL-based methods fail to produce high-quality solutions (Proposition 1), which motivated us to combine CL and WCE loss. Our extended loss function has shown to significantly improve both feasibility and solution quality compared to pure CL-based methods.

---

### Meta-Review · Area_Chair_9WKM · 2026-01-07

**Summary:**

The paper proposes an ML-guided primal heuristic for Mixed Binary Quadratic Programs (MBQPs) utilizing a custom tripartite graph neural network and a combined Weighted Cross-Entropy and Contrastive Learning loss function. Part of the concerns are addressed, including the lack of relevant baselines, which the authors resolved by adding comparisons to the self-supervised method Surco and the commercial solver Gurobi, as well as clarifying solution counts for their data collection procedure. However, numerous concerns are not addressed, including the request for a quantitative cost-benefit accounting of the expensive training process, the lack of systematic Pareto analysis for scalability and dense/near-infeasible regimes, the omission of diverse benchmarks like QPLIB, and an inconsistent justification for excluding competing graph representations based on theoretical limitations that also apply to their own method.

**Reviewer Concerns:**

### [urQJ]

Question 1: Not fully addressed. The reviewer asked: “Why does the CL-based method fail to produce feasible solutions for QMKP, CQKP, and WFLOP but perform well on CBQP?” Only the first part is addressed. The authors point to Proposition 1, which argues that the CL loss fails to supervise variables that are “stable” (i.e., take the same value) across positive and negative samples. This provides a general mechanism for why CL can fail. However, they do not explain why the failure manifests on some benchmarks but not on CBQP. The reviewer was specifically asking what makes CBQP different, and likely expected supporting metrics/statistics showing CBQP is unique in some way.

Other concerns are addressed.

### [g6qN]

Weakness 1 is addressed.

Weakness 2: Not fully addressed. The rebuttal acknowledges that training-data collection is expensive and argues this cost is acceptable because it is incurred offline, and it adds a performance comparison to a self-supervised baseline (Surco) to suggest the extra supervision cost buys better primal integral. However, it does not provide the cost–benefit accounting the reviewer requested: there is no quantitative breakdown of data-collection/training cost, no amortization or “per-solve” cost comparison, and no discussion of maintenance/retraining frequency in practice. As a result, the concern is only partially addressed rather than fully resolved.

Weakness 3: Not addressed. The AC cannot allocate the authors' discussions on feasibility control and charts characterizing failure modes for "dense/near-infeasible cases."

Weakness 4: Not addressed. While the authors briefly mention that their method works on the WFLOP benchmark (which has a slightly higher quadratic density of 32.42% compared to the 25% in synthetic benchmarks), they fail to provide the systematic analysis requested. In particular, The authors explicitly admit they have not yet performed these experiments, stating only that they "plan to include experiments with larger instances for the final version". Moreover, there is no new data or analysis presented regarding condition numbers or a systematic sweep of quadratic densities to show trade-offs (Pareto analysis).

Question 1: Not fully addressed. The authors do provide a new table showing feasibility rates for the ML methods with $p=1.0$ (i.e., using the full model prediction without refinement). However, the reviewer asked for a more thorough ablation across different densities and near-infeasible regimes, as well as a mechanism-level breakdown of how much each component reduces infeasibility. They also do not clearly explain what they would do if infeasibility remains high, beyond adjusting $p$.

Question 2: See "Weakness 2"

### [LE28]

Weakness 1 is addressed.

Weakness 2: Not addressed. The authors do not provide results on QPLIB (or other benchmarks) as requested.

Question 1: Not addressed. While the authors argue that they did not use the representation proposed in (Chen, et al. 2024) for this submission because that representation can not universally represent general QPs with discrete variables in theory. However, they still do not provide the empirical ablation the reviewer asked for. Moreover, this theoretical argument is not fully convincing as a differentiator, since their adopted tripartite GNN is also message-passing based, and its expressive power is likewise subject to WL-type limitations.

### [n1H8]

All concerns are addressed.

**Reviewer Scores:**

The AC anticipates the following outcomes:

Reviewers urQJ and LE28 are likely to maintain their scores (6 and 4) since their concerns were only partially addressed.

Reviewer g6qN may lower their score from 4 to 2, as the majority of their issues remain unresolved.

Reviewer n1H8 is expected to raise their score from 4 to 6, as their concerns have been all satisfied.

---

### Decision · Program_Chairs · 2026-01-26

Reject